# Metabolome and Transcriptome Analyses Reveal Flower Color Differentiation Mechanisms in Various *Sophora japonica* L. Petal Types

**DOI:** 10.3390/biology12121466

**Published:** 2023-11-25

**Authors:** Lingshan Guan, Jinshi Liu, Ruilong Wang, Yanjuan Mu, Tao Sun, Lili Wang, Yunchao Zhao, Nana Zhu, Xinyue Ji, Yizeng Lu, Yan Wang

**Affiliations:** 1Key Laboratory of National Forestry and Grassland Administration on Conservation and Utilization of Warm Temperate Zone Forest and Grass Germplasm Resources, Shandong Provincial Center of Forest and Grass Germplasm Resources, Jinan 250102, China; 2College of Forestry, Shandong Agricultural University, Tai’an 271018, China; 3State-Owned Yishan Forest Farm in Yishui County, Linyi 276400, China

**Keywords:** *Sophora japonica*, anthocyanin, metabolome, transcriptome

## Abstract

**Simple Summary:**

Flower color is one of the most important ornamental characteristics of *Sophora japonica*. The most common *S. japonica* flower color is generally yellow and white, with no exceptional characteristics. The monotonous flower color seriously affects the ornamental value of this species. Therefore, the elucidation of the underlying biosynthetic mechanisms and the role of anthocyanins in the development and cultivation of new flower color varieties in *S. japonica* is of significant importance but has remained unclear to date. In this study, the *S. japonica* ‘AM’ mutant was investigated due to its flower color mutation. The key structural genes involved in the synthesis and accumulation of anthocyanins in *S. japonica* ‘AM’ were screened out using metabolomics and transcriptomics approaches. Bioinformatics analysis demonstrated that the primary key metabolites that affect the color variation in ‘AM’ petals are cyanidin-type anthocyanins. The late genes and the *bHLH* transcription factor are the key genes affecting the anthocyanin synthesis and responsible for color variation in *S. japonica* ‘AM’ petals. This study analyzed the variation patterns and the synthesis pathways of anthocyanin components in *S. japonica* petals. The major relevant regulatory genes were also explored to provide a theoretical basis for the innovative utilization of *S. japonica* flower color germplasm resources.

**Abstract:**

*Sophora japonica* L. is an important landscaping and ornamental tree species throughout southern and northern parts of China. The most common color of *S. japonica* petals is yellow and white. In this study, *S. japonica* flower color mutants with yellow and white flag petals and light purple-red wing and keel petals were used for transcriptomics and metabolomics analyses. To investigate the underlying mechanisms of flower color variation in *S. japonica* ‘AM’ mutant, 36 anthocyanin metabolites were screened in the anthocyanin-targeting metabolome. The results demonstrated that cyanidins such as cyanidin-3-O-glucoside and cyanidin-3-O-rutinoside in the ‘AM’ mutant were the key metabolites responsible for the red color of the wing and keel petals. Transcriptome sequencing and differentially expressed gene (DEG) analysis identified the key structural genes and transcription factors related to anthocyanin biosynthesis. Among these, *F3′5′H*, *ANS*, *UFGT79B1*, *bHLH*, and *WRKY* expression was significantly correlated with the cyanidin-type anthocyanins (key regulatory factors affecting anthocyanin biosynthesis) in the flag, wing, and keel petals in *S. japonica* at various flower development stages.

## 1. Introduction

*Sophora japonica* L. is widely distributed throughout the northern and southern parts of China. Due to its medicinal, timber, and ornamental economic value, it is often used not only as street trees and courtyard tree species but also in landscaping [1,2]. In addition, *S. japonica* has a significant ecological value including improving microclimate and regulating humidity and temperature in the cultivation area. *S. japonica* is also cold-resistant and salt-alkali resistant, and can absorb a large amount of harmful gases in the air to improve air quality [3]. It is of great significance therefore to perform research related to this species to promote the development and utilization of its germplasm resources.

Flower color is one of the primary traits contributing to the ornamental quality of garden landscape plants, and it is also the most adaptable plant phenotypic trait during the evolutionary process [4,5]. The major pigments affecting flower color are flavonoids, carotenoids, and alkaloids [6]. Flavonoids are the main pigment group involved in plant flower color formation including anthocyanin and anthochlor. In most plants, the variation in flower color is due to anthocyanin type and content. Anthocyanins are composed of aglycones and sugars. The aglycones have a circular basic carbon skeleton structure of C_6_-C_3_-C_6_. Naturally existing aglycones are primarily present in the three types of anthocyanins: delphinidin, pelargonidin, and cyanidin. Peonidin is formed by 3’ hydroxy methylation of cyanidin, petunidin is formed by 3’ hydroxy methylation of delphinidin, and malvidin is formed by simultaneous methylation of 3’ and 5’ hydroxy groups of delphinium [7]. The chemical property of aglycone is unstable and generally does not exist in a free state. It often condenses with glucose, rhamnose, and galactose to form stable anthocyanins [8,9]. Anthocyanin content in petals will affect the hue of the flower color with increased content making the color darker and red or purplish red. In addition, anthocyanin content not only affects the color of petals to attract pollinators but also improves the plant’s resistance to abiotic stresses [10,11]. Furthermore, related studies have shown that anthocyanins also play a positive role in the prevention of cancer, cardiovascular diseases, and some chronic diseases of the human body [12,13,14].

The anthocyanin biosynthesis pathway has been thoroughly studied, including the structural and regulatory genes. The structural genes are commonly present in different tree species, which encode the enzymes affecting anthocyanin biosynthesis pathways. Structural genes are divided into early (such as *CHS*, *CHI*, *F3H*, and *F3′H*) and late biosynthesis genes (such as *F3′5′H*, *DFR*, *ANS*, and *UFGT*) [15,16,17]. Regulatory genes affect the anthocyanin biosynthesis pathway by encoding transcription factor (TF) genes to activate corresponding cis-responsive elements on the structural gene promoter, affecting the expression of structural genes. Furthermore, light, temperature, water, and other abiotic factors can also affect anthocyanin biosynthesis [18,19,20]. The elucidation of the anthocyanin biosynthesis mechanisms and their roles in color development in flowers using genetic engineering technology to carry out flower-color-oriented breeding is one of the most important development directions for cultivating new and specific *S. japonica* varieties [4,21,22].

In recent years, transcriptome and metabolome sequencing have often been used to reveal differentially expressed genes (DEGs), key TF, and biosynthetic mechanisms related to the development of specific traits during plant growth and development. New color *Sophora japonica* cultivars were cultivated such as ‘Hongfen 2’, ‘Huanjian’, etc. In addition, Guo [22] used transcriptome sequencing to study the color differences of *Sophora japonica* L. In this study, transcriptome-sequencing analysis was performed on various *S. japonica* petals with a color mutation at different flower development stages. Anthocyanin component variation patterns and their synthesis pathways were also analyzed. The relevant major regulatory genes were then identified, allowing for further flower color breeding using genetic engineering technology and providing new ideas for new flower and color cultivation of *S. japonica* varieties.

## 2. Methods

### 2.1. Plant Material

The flower color mutant *S. japonica* ‘AM’ petals (yellow-white flag petals and light purple-red wing and keel petals) were cultivated in the Zaoyuan Conservation Bank of the Shandong Provincial Center of Forest and Grass Germplasm Resources. The soil of the planting site is loam, neutral to slightly alkaline. The measured soil organic matter content was 7.53 g/kg, available nitrogen, phosphorus, and potassium were 21.02 mg/kg, 12.54 mg/kg, and 85.27 mg/kg, exchangeable calcium and magnesium were 4.98 g/kg and 0.42 g/kg, and the total available copper, zinc, iron, and manganese content was 16.27 mg/kg. The annual average temperature is 12.9 °C, the extreme maximum temperature is 41 °C, the extreme minimum temperature is −24.5 °C, and the average annual temperature is greater than the accumulated temperature of 4584 °C, which is greater than or equal to 10 °C. The frost-free period is 192 days a year, and the average annual sunshine hours are 2638.5 h. During the same period of rain and heat, the average precipitation was 628 mm, mainly concentrated in June–August. Petals were collected from 5 to 20 July 2022, in the middle of the *S. japonica* ‘AM’ tree crown, at various developmental stages: the flower bud stage (S1, four days after dew), initial flowering stage (S2, the flag petals were slightly expanded), the full flowering stage (S3, the third day after the petals were fully expanded), the final flowering stage (S4, the flag petals were decayed and the wing petals were slightly yellowed), and the four flower development stages of each flower development stage. The three types of petals that were collected for this study included the flag (QB), wing (YB), and keel (LGB) (Figure 1). A total of 36 samples (4 flower development stages, 3 petal types, 3 biological replicates each) were collected. Three biological replicates were collected from different branches on the mutated individual plant with consistent growth conditions.

### 2.2. Color Difference Value Determination of S. japonica Petals

A colorimeter was used to measure the color difference values of the four flower development stages (S1–S4) of *S. japonica* ‘AM’ flowers and the three petal types QB, YB, and LGB. Other measurements collected included the brightness value (L*), the redness and greenness (a*), and the yellow-blue degree (b*). Each petal type was randomly selected for measurement at three positions and measured three times at the four flower development stages.

### 2.3. RNA Extraction and Library Construction

The total RNA of the ‘AM’ petals was extracted using a Plant RNA Kit (Omega, R6827). The library construction included the following steps: RNA quality checking (Nano-Drop-2000, Thermo, Waltham, MA, USA; RNA Nano 6000 Assay Kit of the Bioanalyzer 2100 system, Agilent Technologies, Santa Clara, CA, USA), library construction (BGI MGIEasy RNA, MGI Tech Co., Ltd., Shenzhen, China), library purification (Ampure XP Beads, Beckman, Brea, CA, USA), quality control verification (Agilent Technologies 2100 bioanalyzer, Agilent Technologies, Santa Clara, CA, USA), sequencing (DNBSEQ (T7), Benagen Technology Co., Ltd., Wuhan, China).

### 2.4. Sequencing Data Filtering and Reference Genome Comparison

The raw sequencing data were filtered by removing reads using the following criteria: (1) reads containing more than 5% N bases, (2) low-quality (quality value ≤ 5) sequences with a base number of 50%, (3) reads containing adapter contamination, and (4) repetitive sequences caused by PCR amplification. The low-quality sequences and adapters were removed from the raw data to obtain high-quality clean reads to ensure the reliability of the results. The clean reads were compared with the *S. japonica* reference genome (NCBI Project number PRJNA814452) using Star2.7.9a software.

### 2.5. DEG and TF Screening

The DEG data of each sample were analyzed using DESeq2 (1.26.0) software. The screening threshold was padj < 0.05 and |log2FoldChange| > 1. When the number of DEGs was too small, the screening threshold was *p*-value < 0.05 and |log2FoldChange| > 1.

### 2.6. Functional Annotation and Enrichment Analysis of DEGs

Selected DEGs were annotated using the Gene Ontology (GO) database. Compared with the background genes, the GO terms with significant enrichment of DEGs (foreground genes) were identified using the hypergeometric distribution test. The selected DEGs were annotated using the Kyoto Encyclopedia of Genes and Genomes (KEGG) database. The proportion of DEG (foreground gene) annotated genes to the screened DEGs, calculated using the hypergeometric distribution test, was significantly larger than the background genes annotated to the proportion from the total background genes.

### 2.7. Verification of Related Gene Transcription Levels Using Real-Time Fluorescence Quantitative PCR (qRT-PCR)

The *ubiquitin* gene was used as an internal reference [22] to verify the relative expression levels of sixteen key anthocyanin biosynthesis genes in each sample, which were screened by DEGs in the *S. japonica* ‘AM’ mutant. The primers were designed using Primer Premier 6.0 (Appendix A). Gene expression was analyzed using the LightCycler 480 System. The qRT-PCR reaction system and program are presented in Appendix A.

### 2.8. Anthocyanin-Related Metabolite Extraction

The ground powder of various petal types of *S. japonica* ‘AM’ was mixed at a 10:1 ratio of liquid (50% methanol aqueous solution containing 0.1% hydrochloric acid) to powder and vortexed for 5 min. The samples were then sonicated in an ultrasonic cleaner for 5 min and centrifuged at 12,000 rpm for 3 min at 4 °C. The supernatant was removed and combined twice and the sample was filtered with a microporous membrane (0.22 μm pore size). The sample was stored in a sample injection bottle for LC-MS/MS (QTRAP6500+) analysis.

### 2.9. Anthocyanin-Related Differential Metabolite Screening

Qualitative analysis of mass spectrometry detection data was carried out by constructing a Metware Database (MWDB) based on standards. The multiple reaction detection mode (MRM) was used for quantitative analysis. Parametric test, nonparametric test combined with principal component analysis (PCA), and partial least square discriminant analysis were used to mine differential metabolites. The screening criteria for mining were as follows: metabolites with foldchange ≥ 2 and foldchange ≤ 0.5 were regarded as metabolites with significant differences.

### 2.10. Statistical Analysis

The 2^−ΔΔct^ method was used to calculate the relative gene expression in each sample and Microsoft Excel was used for data visualization. Heat map and cluster analysis were drawn using R-heatmaply (1.2.1) and TBtools, respectively. Cytoscape 3.6.0 was used to draw a correlation network diagram of metabolites and related genes. IBMSPSS26 was used to analyze the significance of the differences. The data in this study are presented as the mean ± standard error (SE) of three replicates.

## 3. Results

### 3.1. Determination of Phenotypic Color Difference Values in S. japonica Petals

From the S1 stage, QB color is different from YB and LGB in the *S. japonica* ‘AM’ mutant. At the same flower development stage, L* and b* of QB were significantly higher than in YB and LGB. However, a* was significantly lower than in YB and LGB, with YB a* being the highest. According to the correlation analysis of the two factors of the color difference value, a* demonstrated a negative correlation with L* and b*, while L* was positively correlated with b* (Appendix A and Appendix A). According to the color difference value analysis between different petal types at the four flower development stages, the a* value of YB was the largest, followed by LGB, with QB as the smallest, and in the initial flowering stage. The a* values of the petals were significantly higher than in the other three flower development stages.

### 3.2. Analysis of Metabolite Components of Different Petal Types at Different Flower Developmental Stages

Target metabolism analysis was performed on three petal types at four developmental stages in *S. japonica* ‘AM’ mutant flowers. A total of 45 anthocyanin-related metabolites were obtained, including 36 anthocyanins and 9 flavonoids. KEGG enrichment analysis demonstrated that these 45 metabolites were primarily enriched in anthocyanin biosynthesis (ko00942), flavones and flavonol biosynthesis (ko00944), flavonoid biosynthesis (ko00941), isoflavone biosynthesis (ko0943), and others. The major metabolic pathway components were 36 types of anthocyanins (80.00%), five types of flavonoids (11.11%), and four types of procyanidins (8.89%).

The content of 45 anthocyanin-related metabolites in the samples was then analyzed. The total content of six of the anthocyanins was higher than 1.5 μg/g. These anthocyanins included cyanidin-3-O-glucoside, cyanidin-3-O-(6-O-p-coumaroyl)-glucoside, cyanidin-3-O-rutinoside, delphinidin-3-O-rutinoside-5-O-glucoside, peonidin-3-O-rutinoside, and peonidin-3-O-glucoside accounting for 60.70–88.12%, 0.52–3.12%, 2.27–6.91%, 0.29–8.62%, 1.30–2.57%, and 2.08–4.81%, respectively, of the total anthocyanin content in each sample. Further analysis demonstrated that cyanidin-3-O-glucoside and cyanidin-3-O-rutinoside content in YB and LGB was significantly higher than in QB, with a pattern of YB > LGB > QB consistent with the results of petal color difference measurements. Therefore, the petal coloration of the ‘AM’ mutant is primarily affected by cyanidin-type anthocyanins such as cyanidin-3-O-glucoside and cyanidin-3-O-rutinoside. The accumulation patterns of anthocyanins in different petal types at the same flower development stage varied (Figure 2).

### 3.3. Analysis of Metabolite Components of Different Petal Types at Different Flower Developmental Stages

The changes in anthocyanin biosynthesis-related metabolites at different flower developmental stages and in different petal types of *S. japonica* ‘AM’ mutant during petal development were elucidated via comparative analyses. In the four flower developmental stages, the number of differential metabolites between the same petal types was minor, with no obvious difference in the types of anthocyanin-related metabolites, indicating that the metabolites in the same petal type were synthesized in the other three after being synthesized in S1. Due to its relative stability during flower development, the petal color was maintained since the metabolic substances degraded slowly.

An analysis of 12 differential combinations of different petal types at the same flower developmental stage demonstrated that the difference in metabolite composition between YB and LGB was small, while the differences in metabolite composition between QB and YB and QB and LGB were large. Therefore, QB was used as the control in further analyses with a focus on eight differential combinations of QB vs. YB and QB vs. LGB in four flower development stages. According to the Venn diagram results, QB vs. YB at S4 contained the largest number of differential metabolites, with a total of 29 species. At each flower developmental stage, the number of differential metabolites between QB vs. YB and QB vs. LGB was relatively large (between 24 and 29). The types of differential metabolites were significantly higher than the differential combinations between the same petal types at different flower developmental stages.

Further Venn diagram analysis was conducted between QB vs. YB and QB vs. LGB in the four flower developmental stages in the ‘AM’ mutant and eight common differences in these combinations were screened out (Figure 3a–d). Differential metabolites were used to identify the key metabolites affecting color variation in *S. japonica* ‘AM’ mutant petals (QB, YB, and LGB). Common differential metabolite analysis was performed on the above eight differential combinations (Figure 3e). QB vs. YB and QB vs. LGB exhibited 15 common differential metabolites in the four flower developmental stages. Cyanidin-type anthocyanins were the most common, with six in total including cyanidin-3-O-glucoside, cyanidin-3-O-xyloside, and cyanidin-3-O-rutinoside. Malvidin and Petunidin were the least common with only one representative each including malvidin-3-O-glucoside and petunidin-3-O-rutinoside, respectively.

### 3.4. KEGG Enrichment Analysis of Differential Metabolites

KEGG enrichment analysis was performed on 15 common differential metabolites screened out from the eight differential combinations of petal types at four flower developmental stages. A total of 12 differential metabolites were enriched in three KEGG metabolic pathways. Anthocyanin biosynthesis (8) contained the largest number of enriched differential metabolites, followed by biosynthesis of secondary metabolites (3), and metabolic pathways (1).

Comparative analysis of the different combinations between the same petal types at different flower developmental stages and between different petal types at the same flower development stage demonstrated that the differential metabolite types screened from the combinations of the same petal types in different flower development stages were relatively large. This may be because although the flower organs are at different developmental cycles, the metabolites are highly stable from the relevant metabolite synthesis (from the bud to the final flowering stage). Furthermore, the differences between different petal types in the same flower developmental stage are relatively small since YB and LGB are similar in color. Therefore, transcriptome analysis primarily focused on the differential combinations of QB vs. YB and QB vs. LGB at each flower developmental stage.

### 3.5. Overview of Transcriptome Sequencing Data of Different Petal Types at Different Flower Developmental Stages

To further investigate the molecular mechanisms of the different colors of QB, YB, and LGB in *S. japonica* ‘AM’, transcriptome sequencing analysis was performed. A total of 36 cDNA libraries were constructed using samples from four different flower developmental stages and three different petal types (three replicates for each sample). A total of 60.314394–66.7322 million clean reads were obtained from the sequencing of each sample. The average Q20 (the average ratio of bases with a base quality value greater than 20) clean base rate was 96.81%. The average Q30 (the average ratio of bases with a base quality value greater than 30) clean base rate was 90.84%. The filtered clean reads were mapped to the *S. japonica* reference genome [23], with an 86.88–95.01% mapping rate. Sequencing data accurately reflected gene expression and were used for further analysis (Appendix A).

### 3.6. Differential Gene Screening and Analysis

According to the set filtering conditions, differential expression analysis was perform and DEGs were screened (Figure 4a). In S1, 3459 (1699 up- and 1760 down-regulated genes) and 1835 DEGs (745 up- and 1090 down-regulated genes) were attained from QB vs. LGB and QB vs. YB, respectively. In S2, the QB vs. LGB and QB vs. YB comparisons identified 3389 (1505 up- and 1884 down-regulated genes) and 1829 DEGs (860 up- and 969 down-regulated genes), respectively. In S3, QB vs. LGB and QB vs. YB comparisons identified 3979 (1979 up- and 2000 down-regulated genes) and 1411 DEGs (640 up- and 771 down-regulated genes), respectively. In S4, 1274 (936 up- and 338 down-regulated genes) and 2026 DEGs (811 up- and 1215 down-regulated) in QB vs. LGB and QB vs. YB, respectively. Based on these results at each stage of flower development in *S. japonica* ‘AM’, the largest number of DEGs was screened from the differential combination of QB and LGB while the lowest number was in YB and LGB.

Venn diagram analysis was performed on the screened DEGs and DEGs common to the four flower developmental stages were screened out. The expression of genes related to each petal type during the various developmental stages in *S. japonica* ‘AM’ mutant was explored via Venn diagram analysis (Figure 4b–e). A total of 1836 and 3390 DEGs were detected at the S1 stage in QB vs. YB and QB vs. GB, respectively, with 1320 common DEGs. In S2, 1830 and 3390 DEGs were detected in QB vs. YB and QB vs. LGB, respectively, with 1290 common DEGs. A total of 1412 and 3980 DEGs were detected at the S3 stage in QB vs. YB and QB vs. LGB, with 1115 shared DEGs. Finally, at the S4 stage in QB vs. YB and QB vs. LGB comparisons, 2027 and 4334 DEGs were detected, respectively, with 1655 shared DEGs. Comprehensive comparison of the DEGs between the different combinations of petal types of different colors at the four developmental stages, a total of 122 common DEGs were screened out (Figure 4f), These genes may affect the flag petals (yellow-white) in *S. japonica* ‘AM’ mutant and be responsible for the color variation in different petal types (i.e., light purple wing and keel petals)

### 3.7. DEG GO and KEGG Enrichment Analysis

GO enrichment analysis of 122 common DEGs demonstrated significant differences in the number of common DEGs enriched in the three GO classifications (Figure 5a). The maximum number of common DEGs enriched in biological processes was 75, including 18 GO terms. The next highest number of common enriched DEGs was 51 in molecular function, including 41 GO terms. Finally, the number of common DEGs enriched in cellular components was the lowest with 16, including 10 GO terms.

In the screening of subclasses related to anthocyanin synthesis, the common DEGs were primarily concentrated in the biological processes of glucuronosyltransferase and UDP-glycosyltransferase activity, cellular composition of vacuole and membrane, and molecular functions of response to auxin and auxin-activated signaling pathways.

To further elucidate the metabolic pathways involved in DEGs of different color petal types at the same developmental stage in the ‘AM’ mutant, the common DEGs were analyzed using KEGG enrichment analysis (Figure 5b). A total of 37 metabolic pathways were enriched, largely focused on the biosynthesis of cofactors (ko01240, 4 DEGs), glycolysis/gluconeogenesis (ko00010, 3 DEGs), glucosinolate biosynthesis (ko00966, 1 DEGs), pantothenate and CoA biosynthesis (ko00770, 2 DEGs), and phenylpropanoid biosynthesis (ko00940, 2 DEGs) related to anthocyanin biosynthesis.

### 3.8. Functional Genes Involved in the Anthocyanin Biosynthesis Pathway

To further identify the functional genes regulating the formation of red petals in *S. japonica* ‘AM’ mutants, the functional genes involved in anthocyanin biosynthesis in three petal types during various developmental stages were analyzed. A total of 28 structural genes involved in anthocyanin biosynthesis were identified, including six encoding *chalcone synthase* (*CHS*), two encoding *chalcone isomerase* (*CHI*), two encoding *dihydroflavonol 4-reductase* (*DFR*) and *anthocyanin synthase* (*ANS*), one encoding *flavonoid 3- hydroxylase* (*F3H*), and *UFGT79B1*. Three genes were also identified to encode *Flavonoid 3′,5′-hydroxylase* (*F3′5′H*).

A positive correlation was detected between *GHAGGene27644* expression and the content of four anthocyanin metabolites, including delphinidin-3-O-rhamnoside and delphinidin-3-O-rutinoside-5-O-glucoside (r = 0.71–0.86). The expression of the gene encoding *ANS* (*GHAGGene01871*) was positively correlated with the content of seven anthocyanin metabolites, including cyanidin-3-O-glucoside (r > 0.71). *UFGT79B1* (*GHAGGene01140*) expression was positively correlated with two anthocyanin metabolites, including peonidin-3-O-rutinoside and peonidin-3-O-glucoside (r = 0.70–0.75) (Appendix A). The metabolic group data analysis demonstrated that red petal development in the ‘AM’ mutant was related to anthocyanin content in petals, such as cyanidin-3-O-glucoside and cyanidin-3-O-rutinoside. Therefore, *SjANS*, *SjF3’H*, and *SjUFGT79B1* were the key structural genes affecting the petal color in the *S. japonica* ‘AM’ mutant.

### 3.9. Analysis of TFs Affecting the Formation of S. japonica

Structural gene expression is commonly affected by TFs during anthocyanin biosynthesis. To further investigate the role of TFs involved in the anthocyanin biosynthesis process in *S. japonica* ‘AM’, a total of 48 TF families were identified among the DEGs. Genes within the *bHLH*, *MYB*, and other families that regulate anthocyanin biosynthesis in a coordinated manner were analyzed. To further explore the relationship between anthocyanin biosynthetic structural genes, TFs, and anthocyanin substances, the highly expressed *bHLH* and *MYB* TFs were screened to construct a related network diagram. The expression levels of late-stage anthocyanin biosynthetic genes in *S. japonica* ‘AM’ mutant, including *SjANS*, *SjDFR*, *SjF3’H*, *SjUFGT79B1*, and others, were highly correlated with anthocyanin substances (Figure 6). Among the six TFs, *SjbHLH1* was identified as a key regulatory gene related to the content of various anthocyanins.

### 3.10. SjbHLH1 Sequence Analysis

Plant phylogeny and homology analyses were performed and a phylogenetic tree was constructed in order to determine the relationship of *SjbHLH1* protein amino acid sequences. The *SjbHLH1* sequences obtained via transcriptome sequencing were compared with BLAST homology in NCBI and the gene sequences with high homology were selected for the phylogenetic tree (Appendix A). *SjbHLH1* protein was shown to be closely related to others in the same leguminous family. This protein was closely related to the ApNAI1-like protein in *Abrus precatorius* L., CcbHLH18 in *Cajanus cajan* L., GmNAI1 in *Glycine max* L., GsbHLH18-like in *Glycine soja*, VuNAI1-like in *Vigna unguiculata*, and VubHLH18 and VuBHLH18-like proteins in *Vigna umbellata*. However, SjbHLH1 was slightly more closely related to the ApbHLH18-like protein in *A. precatorius*, the LjHLH18-like protein in *Lotus japonicus*, and VrbHLH18 in *Vigna radiata*.

### 3.11. Anthocyanin Biosynthesis Pathway

The anthocyanin biosynthesis pathway in *S. japonica* ‘AM’ mutant was predicted according to the differential metabolites and DEGs. L-phenylalanine was used as the precursor of the phenylpropane biosynthesis pathway (Figure 7) to form 4-coumaryl-CoA under the catalysis of PAL, C4H, 4CL, and other enzymes, and then it entered the flavonoid biosynthesis pathway. *PAL* (*GHAGGene09401* and *GHAGGene19397*) was highly expressed during S1, with the highest expression identified in S1QB. *4CL* (*GHAGGene23584*) exhibited low expression in both the S1 and S2 stages of flower development with expression in YB and LGB and the S3 and S4 stages of flower development with expression higher in YB than in LGB.

4-coumaryl-CoA was continuously involved in anthocyanin biosynthesis throughout naringenin and eriodictyol synthesis. There were two ways to synthesize eriodictyol in the ‘AM’ mutant. The first was that 4-coumaryl-CoA synthesized naringenin under the action of CHS, CHI, and other enzymes. The second strategy used was that 4-coumaryl-CoA was synthesized by chlorogenic ester as an intermediate product under the action of C3′H and HCT. In the first biosynthesis pathway, *CHS* (*GHAGGene19992* and *GHAGGene04222*) was highly expressed during S3 in YB and LGB, *CHI* (*GHAGGene13892*) was highly expressed during S1 and S2, specifically in YB and LGB, and *F3′H* (*novel1705*) was highly expressed during S1, specifically in YB. In the second biosynthesis pathway, *C3′H* (*GHAGGene04024*) was highly expressed during S3 in QB and LGB and *HCT* (*GHAGGene29329* and *GHAGGene29330*) was highly expressed in YB during S1 and S2 stages.

Eriodictyol was detected as an important intermediate in cyanidin and peonidin synthesis. Three types of dihydroflavonols (i.e., dihydromyricetin, dihydrokaempferol, and dihydroquercetin) were synthesized from pentahydroxy flavone, naringenin, and eriodictyol in the presence of F3H. Dihydrokaempferol synthesized dihydroquercetin under the action of F3′5H and quercetin entered flavones and flavonol biosynthesis pathways under the action of FLS, competing with anthocyanin biosynthesis. The above three types of dihydroflavonols formed corresponding colorless anthocyanins under DFR catalysis and then synthesized colored anthocyanins under the action of ANS. Finally, anthocyanins formed stable anthocyanins with glycosidic bonds under the action of glycosyltransferase. Among them, *F3H* (*GHAGGene06185*) was relatively highly expressed during the S1 stage in QB and YB, *F3′5′H* (*GHAGGene03412* and *GHAGGene29358*) was highly expressed in LGB during all examined developmental stages, *FLS* (*GHAGGene21877*) was highly expressed in YB during all examined developmental stages, *DFR* (*GHAGGene00696*) was expressed only during S2 with the highest expression in QB, and *ANS* (*GHAGGene29358*) was expressed in YB and LGB during all examined developmental stages with highest expression during S2 (especially in YB).

Based on the analysis of the expression patterns of key genes involved in the anthocyanin biosynthesis pathway and the analysis of anthocyanin metabolites of the three petal types during the four developmental stages in *S. japonica* ‘AM’ mutant, two primary rate-limiting enzymes were detected. These enzymes (upstream CHI and downstream ANS) positively regulated anthocyanin synthesis in the petal anthocyanin biosynthesis pathway. CHI activity affected the intermediate product of anthocyanin synthesis and ANS was the key enzyme for the formation of color in the colorless anthocyanins. The decrease in ANS activity directly led to the lighter color of petals or no specific color.

In addition to direct regulation via structural genes, TFs can also indirectly regulate anthocyanin biosynthesis by promoting or inhibiting the structural gene expression. Among the 122 screened DEGs, TF expression patterns of *bHLH*, *MYB*, and *WRKY* affecting anthocyanin synthesis were analyzed. Combined with the results of metabolic group content determination, *bHLH* and *WRKY* were shown to positively regulate anthocyanin biosynthesis by promoting *ANS* and *4CL* gene expression, respectively. However, *MYB* inhibited *CHI* expression and negatively regulated anthocyanin biosynthesis.

### 3.12. qRT-PCR Verification of Key Genes

Transcriptome sequencing data reliability was verified using qRT-PCR using the flag petals (S1QB) of the ‘AM’ mutant as a reference. These genes include early genes for anthocyanin biosynthesis, late genes for anthocyanin biosynthesis, and key TFs. The results demonstrated that the expression trend of the 16 selected genes in each sample was consistent with the RNA-seq expression levels (Figure 8).

## 4. Discussion

Anthocyanins are important secondary metabolites in plants and play a vital role in their growth and development. Anthocyanin biosynthesis is a topic of interest in plant secondary metabolite research [24,25,26]. As a water-soluble pigment, it is usually synthesized in the cytoplasm and endoplasmic reticulum membrane of plants, then transported to various organs with the help of transport proteins, and stored in organelles such as vacuoles and cell walls [27].

In *S. japonica* ‘AM’ flowers, the various petal types (flag, wing, and keel) exhibit different colors. The flag petal in the flower is yellow-white, while the wing and keel petals are light purple-red (Figure 1). The mechanism of petal color development is not yet understood. Through combined analyses of the chemical and transcriptome, Wu et al. [28] found out that the anthocyanins that affect the red petals of *Rosa rugosa* are cyanidin 3,5-O-diglucoside (Cy3G5G) and peonidin 3,5-O-diglucoside (Pn3G5G), and candidate key transcription factors identified via correlation analysis, *RrMYB108*, *RrC1*, and *RrMYB114*, might play critical roles in the control of petal color by regulating the expression of both *RrAOMT* and other multiple structural genes. In addition, a combined analysis of the metabolome and transcriptome in red and white *Bauhinia variegata* cultivars found that d delphinidin 3-O-galactoside and petunidin 3-O-galactoside accumulated the highest in red petals, and the expression of anthocyanidin *BvANR* and *BvF3′5′H* was also the highest [29]. Transcriptomics and metabolomics analyses can elucidate the underlying mechanisms of the color variation, further enrich the molecular biological mechanisms of plant tissue and organ coloration, and provide a theoretical basis for the innovative utilization of *S. japonica* germplasm resources.

### 4.1. Cyanidin-3-O-Glucoside Is the Key Metabolic Substance Affecting S. japonica ‘AM‘ Petal Variation

Anthocyanins are one of the most common natural pigments in plants that affect the color of plant tissues. Specifically, anthocyanins are responsible for the purple in asparagus (*Asparagus officinalis* L.) [30], black seeds in peanut (*Arachis hypogaea* L.) [31], pink in cotton (*Gossypium hirsutum* L.) [32], and other colors in ornamental flowers including *Primula malacoides* Franch. [33], *Muscari botryoides* Mill [34], *Camellia japonica* L. [35], and many others. The specific expression or uneven distribution of anthocyanins in plant tissues is also the primary cause of color variation in flowers [36].

Anthocyanin type and content will affect plant color. These pigments can be divided into six general categories: cyanidin (red to pink), pelargonidin (orange to brick red), delphinidin (blue to blue-purple), peonidin (purple to blue-purple), petunidin (purple to blue-purple), and malvidin (purple to blue-purple) [37]. Furthermore, the pH value in the vacuole, ambient temperature, and UVB can also affect anthocyanin color [5,24,38]. Anthocyanins rarely exist in monomer form and often combine with glycosides to form anthocyanins.

In this study, the metabolic groups in the petals of the flower color mutant *S. japonica* ‘AM’ were determined. A total of 36 anthocyanin metabolites were detected from three petal types during four developmental stages. These metabolites included six glycosides, which primarily existed in the 3-position in the form of monoglycosides, while diglycoside replaced the 3-and 5-hydroxy position. This may be related to the stable physical and chemical properties of anthocyanins [32,39]. During each flower developmental stage in the ‘AM’ mutant, four key anthocyanin metabolites were detected, which affected the color variation between QB, YB, and LGB petals. These included cyanidin-3-O-glucoside, cyanidin-3-O-rutinoside, peonidin-3-O-rutinoside, and peonidin-3-O-glucoside. The anthocyanins which exhibited the greatest influence on the petal color were cyanidins such as cyanidin-3-O-glucoside and cyanidin-3-O-rutinoside. The content of these four anthocyanins in YB and LGB petals that were light purplish red was significantly higher than in QB petals that were yellow and white. Cyanidin-3-O-glucoside was the most abundant anthocyanin, with its content the highest during the S2 stage. This result was consistent with the trend of petal color variation measurements. Therefore, cyanidin-3-O-glucoside accumulation was hypothesized to be the key factor for the color variation of different petal types in *S. japonica* ’AM‘ mutant. This was consistent with previous studies on camellia [35,40] and cotton [41]. In many other plants, including *Prunus salicina* Lindl. [42], *Rhododendron simsii* Planch. [43], *Malus pumila* Mill. [44], and others, cyanidin is also responsible for the red petal color.

In addition, the anthocyanins of delphinidin were detected in *S. japonica* ‘AM’ petals for the first time. Among these, delphinidin-3-O-sophoroside and delphinidin-3-O-arabinoside content in QB was higher than in YB and LGB during the S1 and S2 stages. However, no content was detected during the S3 and S4 stages, suggesting that these two kinds of delphinidins degraded quickly following synthesis during the first two stages due to difficulty in accumulation during flower development. Delphinidin-3-O-rutinoside-5-O-glucoside was present in QB, YB, and LGB during all examined developmental stages, but did not display the blue color in QB and did not affect YB and LGB color. This may be related to the low content of delphinidin metabolites and other biological conditions, such as pH value, sunlight, temperature, etc., that affect the coloration of delphinidin in *S. japonica* petals. The related color-rendering mechanism requires further investigation.

### 4.2. Structural Gene Expression Analysis of Petal Color in S. japonica

Anthocyanin biosynthesis is a branch of the flavonoid biosynthesis pathway. This pathway has been thoroughly investigated in apple (*Malus pumila* Mill.), *Arabidopsis thaliana* L., grape (*Vitis vinifera* L.), and other species [6]. Anthocyanin biosynthesis uses phenylalanine as the direct precursor. Anthocyanins are synthesized by a series of enzymatic reactions such as glycosylation, methylation, acylation, and hydroxylation, which are then transported to various organs by transporters and stored in the vacuoles and cell walls [35,45]. Anthocyanin biosynthesis in higher plants requires the coordinated participation of a variety of enzymes. Among these, *CHS* regulates the continuous decarboxylation and condensation reaction with acyl-CoA as the precursor to form a polyketone intermediate, which provides a basic carbon skeleton for anthocyanin synthesis [46]. *CHS* expression directly affects anthocyanin synthesis. For example, *CHS* down-regulation leads to changes in *Petunia hybrida* Vilm [47] and begonia (*Malus spectabilis* (Ait.) Borkh) flower color [48]. Polychromatic flowers [49] with a red sector on a light-colored background can be obtained by carrying the *CHI* gene in petunia. *F3′5′H* is a key gene that regulates the synthesis of paeoniflorin [50] and catalyzes the 3′5′ hydroxylation of dihydrokaempferol and combines with the corresponding sugars to form anthocyanins. When *F3′5′H* cannot perform the coding function normally, the petals appear pink, but when this gene functions normally, the petals appear purple [51]. *ANS* is an important gene in the downstream structural genes of the anthocyanin biosynthesis pathway. Anthocyanin synthase is a glutarate-dependent oxygenase that can catalyze colorless anthocyanins to synthesize colored anthocyanins. The product is the first chromogenic compound in the anthocyanin biosynthesis pathway [52]. Huang et al. [53] transformed the tulip (*Tulipa gesneriana* L.) *TgANS* gene into tobacco. The transgenic flowers turned red with significantly increased anthocyanin content. However, the deletion or non-expression of *ANS* leads to colorless or bright plant tissues and organs. The deletion of the nucleic acid site in the *SmFAS* gene in eggplant (*Solanum melongena* L.) leads to the premature termination of the *ANS* gene and the formation of plants with white flowers [54].

In this study, the expression of key structural genes in the anthocyanin biosynthesis pathway in various petal types during different flower developmental stages was analyzed. The results of RNA-seq sequencing and qRT-PCR demonstrated that the expression of *SjANS*, *SjUFGT79B1,* and *SjbHLH1* genes in all examined developmental stages exhibited a trend of YB > LGB > QB. This was consistent with the varying trend of cyanidin-3-O-glucoside, cyanidin-3-O-rutinoside, and peonidin-3-O-rutinoside content in the metabolic group. High expression of *SjANS*, *SjUFGT79B1*, *SJbHLH1*, and other genes is, therefore, the primary reason for the synthesis and accumulation of anthocyanins, affecting the color variation of different petal types. These results are consistent with other studies in rose (*Rosa rugosa* Thunb) [55], freesia (*Fressia hybrida* Klatt.) [56], and strawberry (*Fragaria ananassa*) flowers [57].

### 4.3. TF Analysis in S. japonica Petals

Anthocyanin biosynthesis is controlled by structural and regulatory genes [58], in which regulatory genes encode TFs and regulate the expression intensity of structural genes by activating the corresponding cis-responsive elements on the structural gene promoter to affect the anthocyanin biosynthesis pathway. The regulatory genes that affect anthocyanin biosynthesis largely include the *MYB*, *bHLH*, *WRKY*, and *WD40* families [45,59,60,61]. In this study, 37 TF types were detected. However, *WD40* TFs were not found. In YB and LGB, five *bHLH* and four *WRKY* were more highly expressed than in QB at all examined developmental stages. The expression of node regulatory genes of *MYB* genes demonstrated an opposite trend, suggesting that *bHLH* and *WRKY* TFs likely participated in the positive regulation of anthocyanin biosynthesis in *S. japonica* petals. *MYB,* however, negatively regulated the anthocyanin biosynthesis pathway.

*R_2_R_3_-MYB* TFs are known to have the greatest influence on the regulation of anthocyanin biosynthesis, which includes both positive and negative regulation. In addition, these TFs can also regulate the transcription of anthocyanin structural genes [62] with *bHLH* and *WD40* forming protein complexes. Li et al. [56] demonstrated that *AaMYB2* in *Anthurium andraeanum* Linden with various bud colors exhibited higher expression in red and purple varieties, but almost no expression in white and green varieties. Tobacco transformation resulted in a significant up-regulation of anthocyanin synthesis structural genes, which was positively correlated with anthocyanin synthesis. Guo et al. [22] found that *SjMYB2* was the key negative regulatory factor in *S. japonica* petals, consistent with the results of this study.

The *bHLH* TF family is also involved in many plant growth and metabolic pathways, abiotic stress, morphogenesis, and other biological processes, which can directly regulate the late anthocyanin synthesis gene (LBG) promoter to promote or inhibit its expression. A previous study on apples demonstrated that *MdWRKY11* positively regulated late biosynthesis genes including *F3H*, *DFR*, *ANS*, and *UFGT* and participated in the anthocyanin biosynthesis pathway [63].

## 5. Conclusions

Based on the transcriptome and metabolic group analyses of the three petal types during four flower developmental stages of *S. japonica* ’AM’ mutant, the key factors affecting color variation were anthocyanins. *F3′5′H*, *ANS*, *UFGT79B1*, *bHLH*, and *WRKY* were the primary regulatory factors involved in anthocyanin biosynthesis in the mutant petals. The petal coloring in ‘AM’ was largely due to the presence of key anthocyanin metabolites such as cyanidin-3-O-glucoside and cyanidin-3-O-rutinoside. These results can further improve the understanding of the underlying molecular mechanism of the petal color variation in *S. japonica* and provide a basis for cultivating new varieties using genetic engineering technology.

## Figures and Tables

**Figure 1 biology-12-01466-f001:**
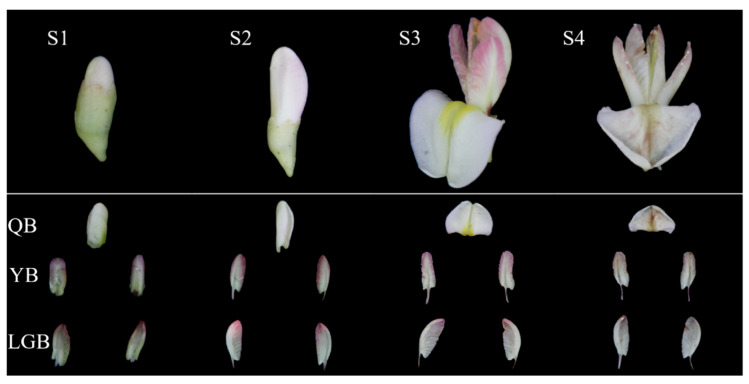
Schematic of petal sample collection of *Sophora japonica* ‘AM’ mutant. S1: Flower bud stage. S2: Initial flowering period. S3: Full flowering stage. S4: Final flowering stage. QB: Flag flap. YB: Wing flap. LGB: Keel flap.

**Figure 2 biology-12-01466-f002:**
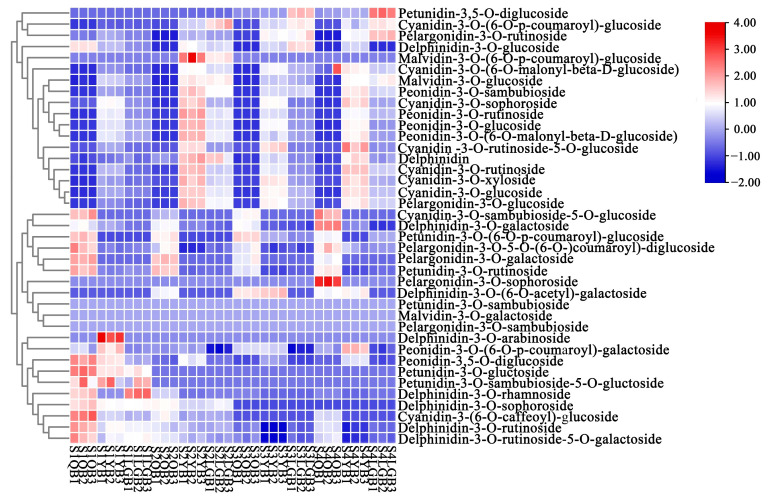
Expression heatmap of metabolites in various samples of the *S. japonica* ‘AM’ mutant.

**Figure 3 biology-12-01466-f003:**
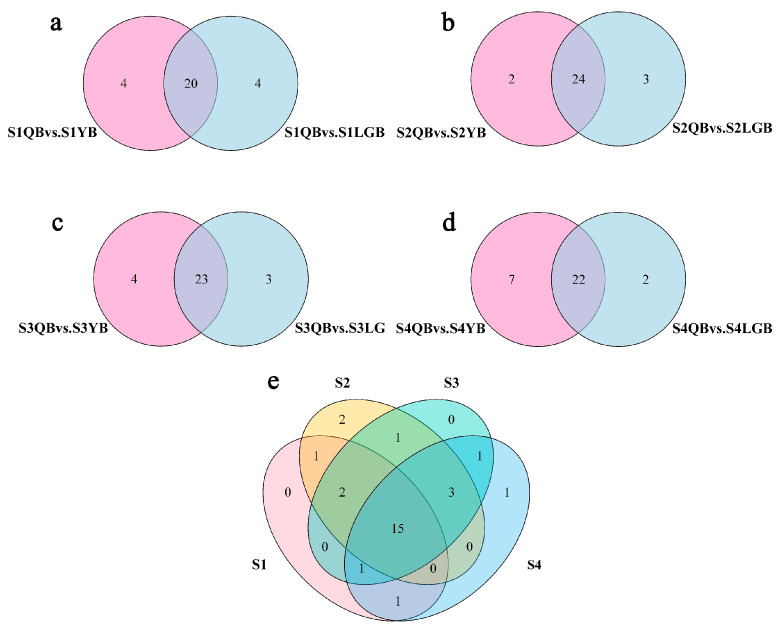
Differential metabolite expression levels in different petal types of *S. japonica* ‘AM’ mutant. (**a**) Differential metabolites of the flag, wing, and keel petals in the bud stage. (**b**) Differential metabolites in the initial flowering period. (**c**) Differential metabolites in the blooming period. (**d**) Differential metabolites in the end flowering period. (**e**) Common differential metabolites at four flower developmental stages.

**Figure 4 biology-12-01466-f004:**
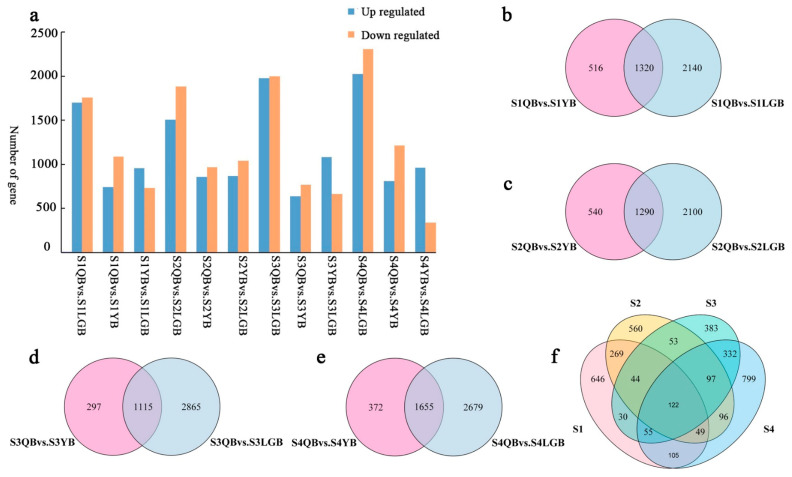
DEGs in each *S. japonica* ‘AM’ sample. (**a**) Number of DEGs in different combinations of petal types during the four flower developmental stages. (**b**) DEGs of the flag, wing, and keel petals in the bud stage. (**c**) DEGs in the initial flowering period. (**d**) DEGs in the blooming period. (**e**) DEGs in the differential metabolites in the blooming period. (**f**) Common DEGs in the four flower developmental stages.

**Figure 5 biology-12-01466-f005:**
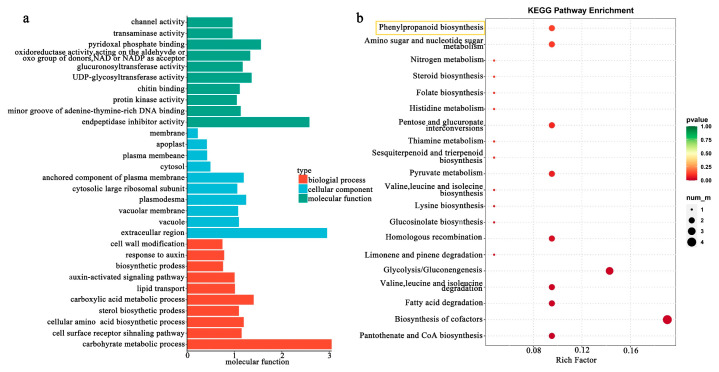
Enrichment of common DEGs in the various petal types at different flower developmental stages of *S. japonica* ‘AM’ mutant. (**a**) GO classifications of DEGs. (**b**) KEGG functional analysis of DEGs. The yellow rectangle is phenylpropanoid biosynthesis (ko00940) in (**b**).

**Figure 6 biology-12-01466-f006:**
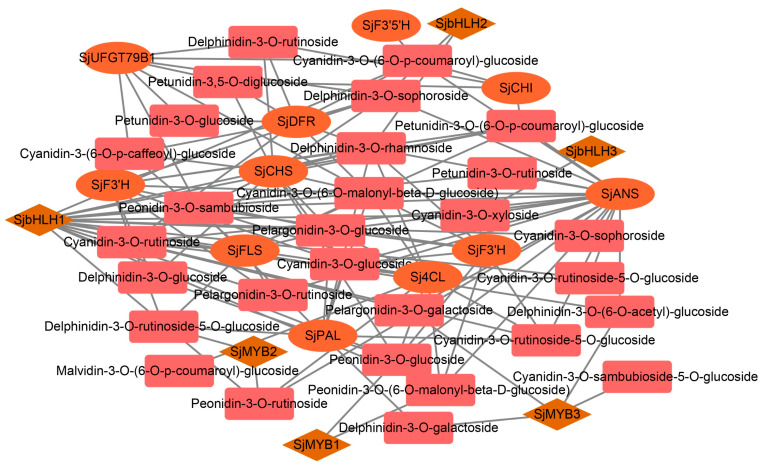
Correlation network diagram of anthocyanin metabolites and related genes in *S. japonica* ‘AM’ mutant. In this figure, the ellipse represents the structural genes, the diamond represents the transcription factors, and the rectangle represents the metabolites.

**Figure 7 biology-12-01466-f007:**
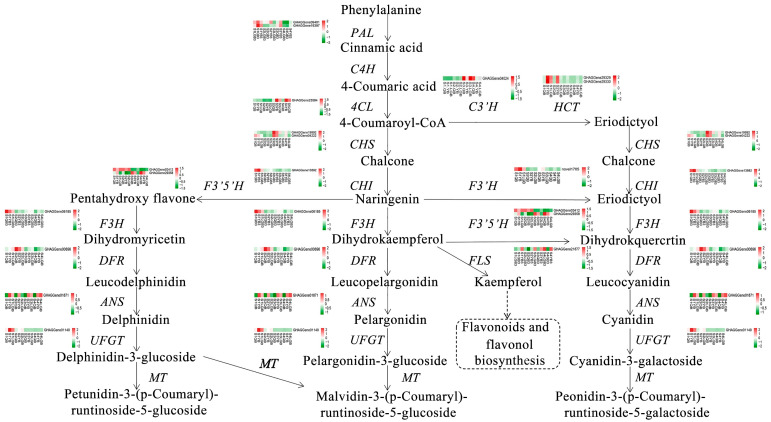
Anthocyanin synthesis pathway of *S. japonica* ‘AM’ mutant petals PAL: Phenylalanine ammonia lyase; C4H: Cinnamic acid-4-hydroxylase; 4CL: 4-coumaryl-CoA ligase; CHS: chalcone synthase; CHI: chalcone isomerase; F3H: flavanone3-hydroxylase; F3′H: flavonoid 3′-hydroxylase; F3′5′H: flavonoid 3′,5’-hydroxylase; DFR: dihydroflavonol 4-reductase; FLS: flavonol synthase; ANS: anthocyanidin synthase; UFGT: UDP-glycose flavonoid glycosyltransferase.

**Figure 8 biology-12-01466-f008:**
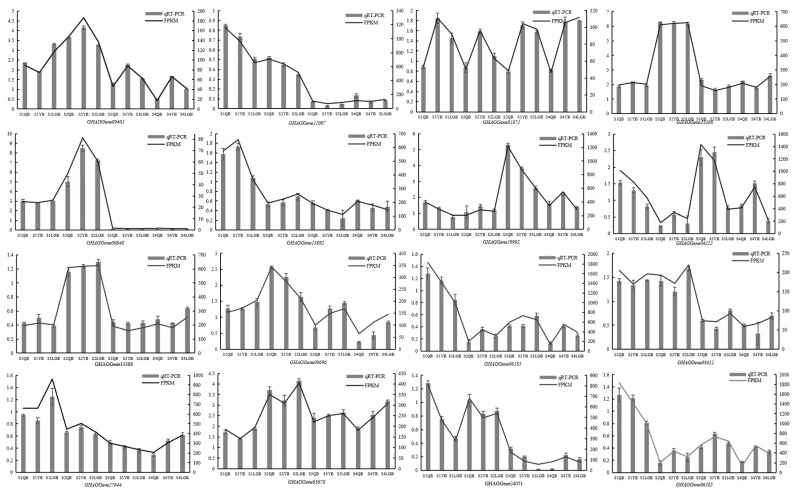
Expression verification of 16 key genes involved in anthocyanin synthesis in S. japonica ‘AM’ mutant petals.

## Data Availability

Data can be provided on suitable request.

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
