# Peer review of "Metabolome and Transcriptome Analyses Reveal Flower Color Differentiation Mechanisms in Various Sophora japonica L. Petal Types"

_biology, 2023, doi:10.3390/biology12121466_

Round 1
Reviewer 1 Report
Comments and Suggestions for Authors
Please see atached file

Reviewer 2 Report
Comments and Suggestions for Authors
Biology-2705404-peer-review-v1
Title: Metabolome and transcriptome analyses reveal flower color differentiation mechanisms in various Sophora japonica L. petal types.
Lingshan Guan1,Jinshi Liu1,Ruilong Wang1,2,Yanjuan Mu1,Tao Sun1,Lili Wang1,Yunchao Zhao1,Nana Zhu3,Xinyue Ji1,Yizeng Lu1*,Yan Wang1.
2. Methods
2.1. Plant material
Figure 1: image must be centered
2.3. RNA extraction and library construction
Nano-Drop-2000: Provide the label of manufactory or Serial number or Country will be good
3. Results
3.2. Analysis of metabolite components of different petal types at different flower developmental stages.
Rephrase this part: including 36 anthocyanins and nine flavonoids. You must to use either numbers or letters like 36 anthocyanins and 9 flavonoids.
3.3. Analysis of metabolite components of different petal types at different flower developmental stages.
Figure 2: At this step, we suggest separating fig2a from the others (fig2 b,c,d,e) for better comprehension, since the writing is close together. The legend should also make it easier to understand.
3.9. Analysis of TFs affecting the formation of S. japonica
Cytoscape as software should be mentioned in the Methods section and not in the Results section.
Question: there are any climate impact affecting S. japonica ‘AM‘ petal variation?
Reviewer 3 Report
Comments and Suggestions for Authors
Dear Author/s,
Your work is good and provides a multitude of information related to the coloration of S japonica flowers through the first of metabolome and transcriptome analyses. In my opinion, I think it is of interest to those in the field of floriculture and not only.
However, some suggestions for improvement are necessary. They are exposed directly in the manuscripts but also here in summary below.
To METHODS
Please explain what the MWDP database is/does? Is this the first time you mention this abbreviation in the work and would it be advisable to detail it? What does it mean? Probably Multiple Weight Decision problem. If so, mention it in a parenthesis.
Moreover, a check is indicated for all the abbreviations in the manuscripts, which must be detailed at the first mention. For all readers of all knowledge levels.
I also suggest detailing the cultivation conditions of the plants from which you collected the samples (petals). Were the cultivation conditions the same (soil, temperature and humidity, degree of pollution) in all repetitions? How was the experimental lot/s organized before the collection of the samples. Were they in the same place? About what area of land did you harvest from and what was the design. So that readers from all over the world understand something about the location, not just the regional ones. Something is not very clear and to strengthen the credibility of the results you should provide more information about the basic biological material.
If you consider it necessary, add a chapter to METHODS
​TO REFERENCES
Check the References and make sure you haven't made too many self-citations. There are many authors/co-authors, 3 of whom are named Wang. I can't tell if the authors/co-authors of the work are listed in the references, because the full names are in the manuscripts, while in the References there are only the initials of the first name..
.......
Otherwise, I am happy with the clarity of the presentation of the results, the conclusions and overall the figures/diagrams/graphs in the additional files are very expressive and hardworking.
Kind regards,

Reviewer 4 Report
Comments and Suggestions for Authors
The manuscript entitled "Metabolome and transcriptome analyses reveal flower color differentiation mechanisms in various Sophora japonica L. petal types" " is an excellently written piece based on well-planned experiments. I have only a few minor comments to add.
Specifically, some figures from the supplement should be relocated to the main text (Figures 4s and 5s- enlarged).
Additionally, some figures need to be enlarged to improve legibility (Figures 2a, f; 3f; 4abc).
At line 120, please include the Agilent kit type.
Line 123:
Please include the sequencing parameters and refer to the software used.
Kindly explain on what grounds the genes for qRT-PCR were selected - literature data or DEG?
Line 497: Kindly provide more detail on the biological conditions.
